# Smoke-Free Home Rules and Association with Child Secondhand Smoke Exposure among Mother–Child Dyad Relationships

**DOI:** 10.3390/ijerph18105256

**Published:** 2021-05-14

**Authors:** Westley L. Fallavollita, Elizabeth K. Do, Julia C. Schechter, Scott H. Kollins, Junfeng (Jim) Zheng, Jian Qin, Rachel L. Maguire, Cathrine Hoyo, Susan K. Murphy, Bernard F. Fuemmeler

**Affiliations:** 1Department of Health Behavior & Policy, Virginia Commonwealth University, Richmond, VA 23284, USA; fallavolliwl@mymail.vcu.edu (W.L.F.); doek@alumni.vcu.edu (E.K.D.); 2Massey Cancer Center, Virginia Commonwealth University, Richmond, VA 23284, USA; 3Department of Psychiatry and Behavioral Sciences, Duke University Medical Center, Durham, NC 27701, USA; julia.schechter@duke.edu (J.C.S.); scott.kollins@duke.edu (S.H.K.); 4Nicholas School of the Environment and Global Health Institute, Duke University, Durham, NC 27708, USA; junfeng.zhang@duke.edu; 5School of Public Health, Guangxi Medical University, Nanning 530021, China; qinjian@gxmu.edu.cn; 6Department of Biological Sciences, North Carolina State University, Raleigh, NC 27695, USA; rachel.maguire@duke.edu (R.L.M.); choyo@ncsu.edu (C.H.); 7Department of Obstetrics and Gynecology, Duke University Medical Center, Durham, NC 27710, USA; susan.murphy@duke.edu

**Keywords:** smoke-free home, smoke-free home rules, child secondhand smoke exposure, salivary cotinine, household smokers, smoking mothers

## Abstract

Smoke-free home rules restrict smoking in the home, but biomarkers of secondhand smoke exposure are needed to help understand the association between smoke-free homes and child secondhand smoke exposure. Participants (*n* = 346) were majority Black/African American mother–child dyads from a longitudinal study in North Carolina. Mothers completed questionnaires on household smoking behaviors and rules, and child saliva samples were assayed for secondhand smoke exposure. Regression models used smoke-free home rules to predict child risk for secondhand smoke exposure. Children in households with smoke-free home rules had less salivary cotinine and risk for secondhand smoke exposure. After controlling for smokers in the household, home smoking rules were not a significant predictor of secondhand smoke exposure. Compared to children in households with no smokers, children in households with at least one smoker but a non-smoking mother (OR 5.35, 95% CI: 2.22, 13.17) and households with at least one smoker including a smoking mother (OR 13.73, 95% CI: 6.06, 33.28) had greater risk for secondhand smoke exposure. Results suggest smoke-free home rules are not sufficient to fully protect children from secondhand smoke exposure, especially in homes with smokers. Future research should focus on how household members who smoke can facilitate the prevention of child secondhand smoke exposure.

## 1. Introduction

Approximately 14 million children ages 3–11 years old are exposed to secondhand smoke in the U.S. each year [1]. Children are uniquely vulnerable to secondhand smoke exposure because they do not have control over their environment and spend most of their time at home [2]. Child secondhand smoke exposure has been causally linked to health consequences including middle ear disease, respiratory symptoms, impaired lung function, lower respiratory illness, and sudden infant death syndrome [3]. Secondhand smoke exposure during childhood is also associated with behavioral problems including poor academic achievement [4] and increased likelihood of cigarette initiation in adolescence [5].

Smoke-free environments are the most effective way to prevent secondhand smoke exposure [3,6]. Smoke-free home rules, which prohibit smoking inside the home, are one such way to reduce child secondhand smoke exposure. Specifically, smoke-free home rules allow for the creation of smoke-free environments [7], where smokers are not willing or able to quit [8]. Additionally, smoke-free home rules help protect children from developing health conditions triggered or exacerbated by smoke [3], becoming smokers themselves [9], and death or injury caused from fires [10]. In the US, most children (89%) live in a household with rules limiting smoking in the home [11]. While North Carolina does not currently require smoke-free rules in homes with children, an estimated 85% of homes with children have voluntarily adopted such rules [11].

Smoke-free home rules may not completely protect children from tobacco smoke exposure. Children who do not live with smokers, or children who live with smokers that do not smoke in the home, can still be exposed to tobacco smoke via dust on household surfaces and the skin and clothing of caretakers [12,13,14]. Parents or caretakers who smoke outside the home but near entrances or windows may also expose their family members to secondhand smoke [15,16]. The risk for secondhand smoke exposure increases further among households with multiple smokers or mothers who smoke [8]. As mothers may be the primary caretakers for children [7,8], children with smoking mothers are especially at risk for secondhand smoke exposure, suggesting mothers are important for assessing a child’s risk of secondhand smoke exposure and as an intervention target [8]. Additionally, it is a challenge to determine true levels of secondhand smoke exposure, as parental reports can conflict with objective measures [17,18].

In Rosen et al. (2015)’s meta-analysis of smoke-free home interventions, households that adopted smoke-free home rules had significant reductions in airborne nicotine and particulate matter (PM), but a small amount remained, suggesting smoke-free home rules are not sufficient to fully protect children [19]. In observational studies, smoke-free home rules have been associated with less reported secondhand smoke exposure [20], lower levels of airborne nicotine [15,21,22], and lower levels of cotinine [21,22,23,24,25]. However, children may have lower levels of cotinine, or live-in homes with reduced airborne nicotine, and still be exposed to secondhand smoke [18,19]. Thus, this study utilizes salivary cotinine, an objective biological measure of smoke exposure [9,26].

The purpose of this study was to examine the association between smoke-free home rules and child secondhand smoke exposure among an understudied population [8]—a community sample of primarily Black/African American mother–child dyads living in North Carolina. According to systematic reviews focused on child secondhand smoke exposure [8,18,19], most published studies examining associations between smoke-free home rules and child secondhand smoke exposure have been conducted on samples outside of the United States [8] or sampled from clinical settings, such as hospitals and primary care settings [18,19]. This study also utilized biochemically validated measures of exposure to nicotine while controlling for the presence of smoke-free home rules and cigarette smokers in the household, as children that live with smokers may experience significant secondhand smoke exposure [8,21]. In conducting this research, we aimed to disentangle the contribution that mothers’ smoking has beyond that of others in the home, within homes that vary with respect to the presence of smoke free home rules. We hypothesized that: (1) the presence of cigarette smokers in the household and the cigarette smoking status of the mother would be associated with secondhand smoke exposure, (2) homes with the presence of cigarette smokers within the household would be less likely to endorse having smoke-free homes, and (3) the presence of smoke-free home rules would be protective against secondhand smoke exposure.

## 2. Materials and Methods

### 2.1. Participants

Participants were drawn from the Newborn Epigenetic Study (NEST), a prospective cohort study of pregnant women examining effects of prenatal exposures on epigenetic profiles and child development. A detailed description of identification and enrollment procedures has been published elsewhere [27,28]. Briefly, between 2005 and 2011, pregnant women were recruited through prenatal clinics serving Duke University Hospital and Durham Regional Hospital Obstetric facilities in Durham, North Carolina. Recruitment occurred in two waves: from 2005–2008, 1101 women were approached to participate and 85% enrolled and from 2009–2011, 2548 women were approached to participate and 67% enrolled. To be included in NEST, women needed to be 18 or older, speak English and/or Spanish, intend to use one of the obstetrics facilities for the index pregnancy, and allow access to labor and birth outcome data. At the time of enrollment, women completed questionnaires which included sociodemographic characteristics (e.g., race/ethnicity, education) and maternal health and lifestyle factors [29,30].

A subset of the mother–child dyads (*n* = 350) returned for a follow-up study, Neurodevelopment and Improving Children’s Health Following Environmental Tobacco Smoke Exposure (NICHES). During follow-up visits (2013–2019), mothers completed expanded questionnaires that asked about household member tobacco use, passive smoke exposure, and home smoking rules, and children provided saliva samples for cotinine analysis. The Institutional Review Board (IRB) at Duke University approved this study (Pro00043781, Pro00014548, and Pro00064859).

This analytic sample was restricted to NICHES participants who provided demographic and tobacco-related data and had a child provide a saliva sample (*n* = 346). NICHES questionnaires were in English only, so there were fewer Hispanic participants (2.6% in current sample vs. 17.8% in full NEST study). Thus, the current sample is not representative of the full NEST cohort. Compared to the full NEST study cohort, mothers in this study were more likely to report their race/ethnicity as Black/African American (58.7% in current sample vs. 43.5% in full NEST sample) and more highly educated (39.6% college graduates in the current sample vs. 30.1% in the full NEST sample). The average age of mothers in this study was 28.1 (SD = 5.7), compared to the full NEST dataset at 28.3 (SD = 5.9).

### 2.2. Collection of Saliva Samples and Cotinine Assay Procedures

Children (mean age = 6.2 years, SD = 2.4) provided saliva samples during at least one NICHES follow-up visit (2013–2019). Saliva samples were typically collected early in the visit (e.g., approximately 15–20 min after arrival). Saliva was assayed for cotinine concentration level (ng/mL) to determine secondhand smoke exposure. Saliva samples were stored in 2 mL tubes at −80 °C before analysis. As described elsewhere [30], assays were completed at the Exposure Biology and Chemistry Lab at Duke University, using high-performance liquid chromatography coupled with tandem mass spectrometry. This assay was designed so that a level of salivary cotinine at 0.05 ng/mL would be detectable at a reproducibility rate > 94%.

### 2.3. Child Secondhand Smoke Exposure

Child secondhand smoke exposure was assessed using cotinine concentration values from saliva samples [17,31,32], Concentration values of ≥1 ng/mL were categorized as secondhand smoke exposed, while values <1 ng/mL were categorized as secondhand smoke non-exposed, as recommended by Benowitz et al. (2011) [26].

### 2.4. Home Smoking Rule

Home smoking rules were based on maternal report using a measure based on Eisner et al. (2001) [33]. When asked how cigarette smoking is handled inside and outside the home, participants who selected “No one is allowed to smoke anywhere inside or outside the home” were considered to have a smoke-free home rule. Participants who selected either remaining option (“Smoking is permitted in some places or at sometimes” and “Smoking is permitted anywhere”) were considered to have no smoke-free home rule.

### 2.5. Other Covariates

Other covariates included the household smoker status (households with no smokers, households with at least one smoker but a non-smoking mother, households with at least one smoker including a smoking mother) maternal race/ethnicity (Black/African American, White/Caucasian, or any other race/ethnicity including Hispanic/Latino), maternal education (college graduate or non-college graduate), household income (<USD 15,000, USD 15,000–USD 30,000, USD 30,000–USD 60,000, >USD 60,000), and child age and gender.

### 2.6. Statistical Analysis

Sample characteristics are reported as frequency counts and percentages. Salivary cotinine levels between households with and without smoke-free home rules were compared using Chi-squared tests, and by household smoker status using Mann-Whitney U and one-way ANOVA tests, at a *p*-value value ≤ 0.05 significance level. Logistic regression models were used to test the association between child risk for secondhand smoke exposure and home smoking rules, controlling for the presence of smokers in the household, to determine if smoke-free home rules were effective at reducing risk for secondhand smoke exposure in households with and without smokers. Fully adjusted models included maternal race/ethnicity, maternal education, household income, home smoking rule, and household smoker status. Additional sensitivity analyses were conducted using the fully adjusted models including maternal race/ethnicity, maternal education, household income, home smoking rule, and household smoker status, but limited to households with single mothers (i.e., mothers who indicated that they were unmarried, divorced, or separated). Analyses were conducted using R 3.6.0 (R Core Team, Vienna, Austria).

## 3. Results

### 3.1. Sample Characteristics

A total of 346 children (mean age = 6.2 years, SD = 2.4) were included in the analysis. Half (52.0%) were female. Among mothers, 58.7% identified as Black/African American and 39.6% were college graduates. While 32.1% of households earned >USD 60,000 per year, 24.3% earned <USD 15,000. There were no smokers in 65.3% of the households and 21.4% of mothers were smokers. Only 58.7% of households had a smoke-free home rule.

As shown in Table 1, a lower percentage of households with no smoke-free home rule had mothers who had graduated from college (22.4% vs. 51.7%, *p* < 0.001), had household incomes of >USD 60,000 per year (20.3% vs. 40.4%, *p* = 0.0013), and no smokers within the household (34.3% vs. 87.2%, *p* < 0.001).

### 3.2. Cotinine Concentration by Home Smoking Rule

Table 2 presents salivary cotinine concentrations by home smoking rule. Overall, children in households with a smoke-free home rule had a significantly lower level of cotinine than children with no rule banning smoking in the home (mean = 0.49 ng/mL vs. 1.76 ng/mL, *p* < 0.001). However, cotinine concentrations were not significantly different between households with and without smoke-free home rules when the household had no smokers (mean = 0.31 ng/mL vs. 0.34 ng/mL, *p* = 0.6215), had at least one smoker and a non-smoking mother (mean = 0.90 ng/mL vs. 1.50 ng/mL, *p* = 0.1018), or had at least one smoker including a smoking mother (mean = 2.35 ng/mL vs. 3.06 ng/mL, *p* = 0.8848).

Within households with a smoke-free home rule, there were significant differences in the level of cotinine associated with the type of smokers in the home. Children in households with at least one smoker, including a smoking mother, had higher cotinine levels than children in households with at least one smoker but a non-smoking mother, and in households with no smokers (mean = 2.35 ng/mL vs. 0.90 ng/mL, vs. 0.31 ng/mL, *p* < 0.001). Similarly, among household with no smoke-free home rule, children in households with at least one smoker, including a smoking mother, also had higher cotinine values than in homes with at least one smoker but a non-smoking mother, and no smokers (mean = 3.06 ng/mL vs. 1.50 ng/mL, vs. 0.34 ng/mL, *p* < 0.001).

### 3.3. Risk for Secondhand Smoke Exposure

Maternal education, household income, and home smoking rule were associated with secondhand smoke exposure, as shown under Model 1 on Table 3. Children with mothers who had graduated from college were less likely to have secondhand smoke exposure compared to children of non-college graduates (OR = 0.31, 95% CI: 0.11, 0.76). Children in household earning USD 30,000–USD 60,000 (OR = 0.32, 95% CI: 0.14, 0.69) and >USD 60,000 (OR = 0.19, 95% CI: 1.90, 6.17) were also less likely to have secondhand smoke exposure than children in households earning <USD 15,000. Children with no smoke-free home rule were more likely to have secondhand smoke exposure compared to children with a smoke-free home rule (OR = 3.52, 95% CI: 1.99, 6.38). After controlling for the presence of smokers in the household (Model 2, Table 3), children in homes with at least one smoker but a non-smoking mother (OR = 5.35, 95% CI: 2.22, 13.17) and children in homes with at least one smoker including a smoking mother (OR = 13.73, 95% CI: 6.06, 33.28) were more likely to have secondhand smoke exposure, compared to children in households with no smokers. In the fully adjusted model, home smoking rule was not a significant predictor of secondhand smoke exposure. We also conducted a sensitivity analysis, including only single mothers (i.e., those who had indicated that they were unmarried, divorced, or separated, *n* = 124). Results demonstrated that associations between having a household smoker (e.g., 1 or more, with non-smoking mother within the home: OR = 11.1, 95% CI: 2.7, 52.9; 1 or more, with smoking mother within the home: OR = 14.5, 95% CI: 4.4, 56.2) and secondhand smoke exposure remained significant.

## 4. Discussion

The purpose of this study was to examine the association between smoke-free home rules and child secondhand smoke exposure among a sample of primarily Black/African American mother–child dyads in North Carolina using salivary cotinine and secondhand smoke exposure cut-off points. Smoke-free home rules were associated with lower levels of salivary cotinine, but not with a reduced risk for secondhand smoke exposure in households with smokers. Results highlight the important role of household members, and especially mothers, in preventing child secondhand smoke exposure, and suggest that mothers are important intervention targets for the protection of young children from secondhand smoke exposure.

Consistent with the existing literature, the presence of smokers in the household was associated with child secondhand smoke exposure [8,18,23,34,35]. Similar to other studies, we found that children in our study who lived with smokers had secondhand smoke exposure, even in households with smoke-free home rules [23,24,25]. This suggests that smoke-free home rules do not fully protect children from secondhand smoke exposure [12,23,24], potentially due to smoke from the outside seeping indoors [12], smokers tracking nicotine indoors [15,16], and children spending time in close proximity to caretakers [36,37]. While this study did not evaluate smoke-free home rules as an intervention, results align with Rosen et al. (2015)’s review which concluded that, given tobacco smoke persists in homes post smoke-free home intervention, population and regulatory measures are needed to fully protect children from tobacco smoke exposure [19].

Despite smoke-free home rules lack of significant association with secondhand smoke-exposure in households with smokers, literature suggests the adoption of smoke-free home rules may be effective at facilitating parental cessation [38,39]. Mills et al. (2009)’s review of smoke-free home rules and adult smoking behavior found that the adoption of smoke-free home rules is associated with increased quit attempts and decreased consumption. Similarly, Collins complex behavioral counseling intervention was successful at increasing bioverified quit rates for smoking mothers and concluded that implementing home smoking restrictions may be an important step in facilitating cessation [38,39]. The adoption of smoke-free home rules may also help reduce or prevent the uptake of alternative tobacco products [40,41]. Given the evidence that smoke-free home rules are associated with less airborne nicotine and lower levels of cotinine, the adoption of smoke-free home rules appears justifiable, especially as part of larger strategies to encourage cessation, but more research is needed.

Households with a smoking mother had the largest association with child secondhand smoke exposure, suggesting that mothers are an important target for interventions to reduce child secondhand smoke exposure. These associations were found across models that included the full sample and those that only included single mothers. Results suggest that the prevention of secondhand smoke exposure should prioritize smoking cessation within homes where mothers smoke, over the sole implementation of smoke free home rules—regardless of marital status. This important finding aligns with prior studies demonstrating maternal smoking behavior is the greatest determinant of child secondhand smoke exposure, compared to other individuals in the household [8,25,35]. Previous studies have found that mothers are important in the adoption of smoke-free home rules: mothers are the household member most likely to instigate smoke-free home rules [42] and children with mothers who smoke are less likely to have a smoke-free home rule, compared to children with fathers who smoke [24]. The perception of health risks associated with smoking may influence a mother’s decision adopt smoke-free rules. For example, mothers who are more aware of the health consequences of smoking are more likely to try to prevent secondhand smoke exposure among their children, and less likely to be smokers themselves [43]. More research is needed to understand how the motivations and smoking behaviors of mothers can be used to develop interventions to promote and enforce smoke-free home rules.

The majority of households (59.0%) in this study had a smoke-free home rule. Households with smokers had lower rates of smoke-free home rules, compared to households without smokers. This selection bias might have biased the present results. Future studies should consider interventional designs that incorporate longitudinal measures, in order to determine the effects that an implementation of a smoke-free home rule among households with smokers on secondhand smoke exposure of children. Our results also showed that a greater proportion of children with mothers who were college graduates lived in smoke-free households. Higher household income was also associated with smoke-free home rules. These findings align with the existing literature [11,23,24,34,44,45,46], which demonstrate that while a majority of households adopt smoke-free home rules [11], rules are less common among homes with smokers [23,24,34,44,45,46], less income [35], and less educational attainment [34,44,47].

Our study demonstrates cross-sectional effects of smoke-free home rules on child secondhand smoke exposure outside the context of intervention. Limitations to our study include the generalizability of the findings given the sample population. Our community sample included a high rate of Black/African American participants (58.7%), is highly educated (39.6%), and received services in North Carolina, the largest tobacco producing state in the United States [48]. Importantly, cotinine concentrations in Black/African Americans are higher compared to Whites at similar levels of smoking [49], so despite this study not finding a significant difference between Black/African American and White study participants’ secondhand smoke exposure, direct comparisons should not be made. It was also not possible to examine the long-term effects of smoke-free home rules on secondhand smoke exposure since salivary cotinine has a half-life of 17.5 h [50]. Additionally, this study did not control for the effects of secondhand smoke exposure from alternative tobacco products, such as cigars and hookah, or electronic nicotine delivery systems (ENDS). We were also unable to determine the possibility of exposure to smoke in cars among children coming to the follow-up visit, prior to saliva collection. Future research could incorporate biological markers that characterize long-term tobacco exposure, such as hair samples [51], collected before and after the implementation of smoke-free home rules, and include expanded questionnaires covering additional potential sources of nicotine exposure, such as e-cigarettes. Finally, there are other variables not measured in the surveys that may be important factors for secondhand smoke exposure that were also not included in the current analyses that could bias the results, such as frequency of tobacco use, paternal education, and air conditioning in the home [8,18,19].

Despite limitations, this study provides insight into which households are likely to implement smoke-free home rules, while emphasizing the important role—over and above home smoking rules—of household members in protecting children from secondhand smoke. It uses biochemically validated measures of secondhand smoke exposure with thresholds based on prior research [26]. Given the limited evidence on effective interventions to protect children from secondhand smoke exposure [19], our study emphasizes the importance of reducing tobacco use by household members and identifies potential areas for prevention and intervention.

Future research should focus on identifying ways that household members who smoke can facilitate the prevention of child secondhand smoke exposure. Exposure to secondhand smoke has been found to cause numerous health problems in children, including more frequent and severe asthma attacks, ear and respiratory infections, as well as increased liability for the initiation and use of cigarettes and increased risk for coronary heart disease, stroke, and cancer in later life [3]. Research areas requiring additional study include behaviors and beliefs that promote the protection of children from secondhand smoke exposure, and smoke-free home rules as an intervention step to encourage caretaker smoking cessation.

## 5. Conclusions

As hypothesized, the presence of cigarette smokers within the household was associated with secondhand smoke exposure in children. Households with smokers, and especially households with smoking mothers, were less likely to have a smoke-free home rule. This also aligns with our initial hypotheses. However, we had not hypothesized that smoke-free home rules would be not sufficient to protect children from secondhand smoke exposure among households with smokers in our sample. These findings suggest that efforts to reduce child secondhand smoke exposure in similar populations focused on encouraging household members to smoke outside of the home environment are likely not effective. The most effective way to protect children from secondhand smoke exposure may be cessation by mothers and other smokers within the household. Though the adoption of smoke-free home rules inside and outside of the home may play an important role in counseling interventions to encourage cessation among household members.

## Figures and Tables

**Table 1 ijerph-18-05256-t001:** Sample characteristics.

	Smoke-Free Rule		No Smoke-Free Rule			Total	
	*n*	%	*n*	%	χ^2^, *p*-Value	*n*	%
Total	203	58.7%	143	41.3%		346	100.0%
Child age					0.29, 0.8667		
3–4	78	38.4%	59	41.3%		137	39.6%
5–6	72	35.5%	48	33.6%		120	34.7%
7–13	53	26.1%	36	25.2%		89	25.7%
Child gender					0.17, 0.6791		
Female	108	53.2%	72	50.3%		180	52.0%
Male	95	46.8%	71	49.7%		166	48.0%
Maternal race/ethnicity					5.18, 0.0750		
White	78	38.4%	42	29.4%		120	34.7%
Black	109	53.7%	94	65.7%		203	58.7%
Other	16	7.9%	7	4.9%		23	6.6%
Maternal education					**29.00, <0.001**		
Non-college graduate	98	48.3%	111	77.6%		209	60.4%
College graduate	105	51.7%	32	22.4%		137	39.6%
Household income					**15.77, 0.0013**		
<USD 15,000	42	20.7%	42	29.4%		84	24.3%
USD 15,000–USD 30,000	40	19.7%	38	26.6%		78	22.5%
USD 30,000–USD 60,000	39	19.2%	34	23.8%		73	21.1%
>USD 60,000	82	40.4%	29	20.3%		111	32.1%
Household smoker status					**104.35, <0.001**		
No smokers in home	177	87.2%	49	34.3%		226	65.3%
1 or more, non-smoking mother	12	5.9%	34	23.8%		46	13.3
1 or more, smoking mother	14	6.9%	60	42.0%		74	21.4%

Bold text indicates statistically significant associations at *p*-value < 0.05.

**Table 2 ijerph-18-05256-t002:** Cotinine concentration by home smoking rule.

	Smoke-Free Rule				No Smoke-Free Rule			
	*n*	Mean	SD	*p*-Value	*n*	Mean	SD	*p*-Value
Total	203	0.49	0.97		143	1.76	2.76	**<0.001**
Household smoker status				**<0.001**				**<0.001**
No smokers	177	0.31	0.63		49	0.34	0.52	0.6215
1 or more, non-smoking mother	12	0.90	1.60		34	1.50	2.38	0.1018
1 or more, smoking mother	14	2.35	1.63		60	3.06	3.39	0.8848

Bold text indicates statistically significant associations at *p*-value < 0.05.

**Table 3 ijerph-18-05256-t003:** Risk for secondhand smoke exposure.

		Model 1		Model 2	
Predictor	*n*	OR	95% CI	OR	95% CI
Maternal race (reference = White)	120				
Black	203	1.17	(0.54, 2.58)	1.75	(0.74, 4.23)
Other	23	1.07	(0.24, 4.09)	1.48	(0.30, 6.36)
Maternal education (reference = Non-college graduate)	209				
College graduate	137	**0.31**	**(0.11, 0.76)**	0.45	(0.15, 1.20)
Household income (reference = <USD 15,000)	84				
USD 15,000–USD 30,000	78	0.58	(0.29, 1.12)	0.69	(0.32, 1.47)
USD 30,000–USD 60,000	73	**0.32**	**(0.14, 0.69)**	**0.42**	(0.17, 1.00)
>USD 60,000	111	**0.10**	**(0.03, 0.31)**	**0.18**	**(0.04, 0.63)**
Home smoking rule (reference = smoke-free home)	203				
No smoke-free home rule	143	**3.52**	**(1.99, 6.38)**	1.20	(0.56, 2.52)
Household smoker status (reference = no smokers)	226				
1 or more, non-smoking mother	46			**5.35**	**(2.22, 13.17)**
1 or more, smoking mother	74			**13.73**	**(6.06, 33.28)**

Model 1 models the risk for secondhand smoke exposure including the predictors of maternal race, maternal education, household income, and home smoking rule. Model 2 models the risk for secondhand smoke exposure including the predictors of Model 1, in addition to household smoker status. Bold text indicates statistically significant associations at *p*-value < 0.05.

## Data Availability

The data that support the findings of this study are available from the corresponding author, B.F.F., upon reasonable request.

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
