# Peer review of "Smoke-Free Home Rules and Association with Child Secondhand Smoke Exposure among Mother–Child Dyad Relationships"

_ijerph, 2021, doi:10.3390/ijerph18105256_

Round 1

Reviewer 1 Report

Thank you for the opportunity to review the revised manuscript, "Smoke-Free Home Rules and Association with Secondhand Smoke Exposure." The study remains timely and important and the authors have addressed the concerns outlined in the prior submission. I do not have additional concerns at the moment and commend the authors on their responsiveness.

Reviewer 2 Report

Revisions prepared by the authors are insufficient and still, sampling methods applied in this study may lead to serious bias. The number of variables analyzed in this study is very limited. There is a limited number of statistics applied and the results are predictable. There is a lack of novel practical implications of this study. 

This study repeats well-known facts. There is a lack of novelty of the results obtained by the authors that result from sampling as well as way of data presentation. Findings from this study are unclear due to the potentially biased sample as well as a limited number of questions applied in this study. The authors should provide a more comprehensive questionnaire (detailed characteristics of household, detailed characteristics of smoking habits) as well as improve sampling methods.

Reviewer 3 Report

The authors did a good job of addressing comments of all reviewers and incorporating feedback into the manuscript. 

Reviewer 4 Report

The authors have provided edits in response to the last round of comments. The literature review is adequate, as is the discussion. Their method is more clearly but could be improved. 

  1. Please clarify the time of saliva samples collection. For example, if they drove to the clinics to do the test, how will they eliminate the possibility of exposure to smoke in car, etc.
  2. The variables are limited. For example, frequency of tobacco use, paternal education, smoking attitudes, air conditioning in the home, etc are important factors of SHS and may bias the results. 
  3. The title should be "Smoke-Free Home Rules and Association with Child Secondhand Smoke Exposure among mother-child dyads relationships" if this study only assessed the association from maternal side. Also would it be better to assess the association among single mothers? 

Reviewer 5 Report

Overall, this is a well written paper with an interesting result on the health area.

INTRODUCTION

The introduction provides a good perspective of the main topic, however, to make the introduction more substantial, the author may wish to provide several actual references.

Motivations for this study are clear and the objectives are clearly defined at the Introduction, the argumentation in this part was concise. I would recommend to include an initial study hypothesis

METHODS

The methodology proposed to reach the aim of the study look appropriate, well designed and conducted.

There are a few instances where assertions are made that are not substantiated with references.

RESULTS

Results paragraph include the most relevant data.

All of the tables include specific, good developed statistic.

DISCUSSION

It would be interesting to include the impact of smoke home in develop of kids and the direct impact in their health

Conclusion should respond concisely the research aim

Round 2

Reviewer 2 Report

This study was significantly improved.

This manuscript is a resubmission of an earlier submission. The following is a list of the peer review reports and author responses from that submission.

Round 1

Reviewer 1 Report

Thank you for the opportunity to review the manuscript, “Smoke-Free Home Rules and Association with Child Secondhand Smoke Exposure.” The manuscript examines whether smoke-free home rules are adequate to fully protect children against secondhand smoke exposure. The study is timely and important. However, some concerns temper my enthusiasm, particularly with regards to generalizability and the potential impact of e-cigarette use on study findings. Specific comments are outlined below.

Major

  • Page 3 Lines 101-108 - Were NEST and NICHES participants compared based on their tobacco use behavior in NEST (or any other variables not noted in the methods)? Does this subsample evidence a relatively higher or lower rate of such use (and thus potentially a heavier or lighter smoking population)?
  • Methods - the most notable methodological limitation is the lack of assessment or differentiation between exposures to SHS from combustible tobacco and potential exposure to nicotine-containing vapor from the use of e-cigarettes which would in most cases (assuming the use of nicotine containing e-liquid or cartridges) also result in positive cotinine on assay.
  • Related, given the higher prevalence of e-cigarette use among a younger cohort of individuals during the assessment period, the authors should both provide descriptive on household e-cigarette use and maternal age.
  • Limitations should note the generalizability of the current findings given the characteristics of the sample, especially in terms of ethnicity and race. In general findings should be contextualized better tot he actual population represented by the sample that was assessed.

Minor

  • Page 3 Line 103 - Change to read “fewer Hispanic participants” rather than “less Hispanic participants.”
  • Were specific follow-ups conducted to compare proportions of smoke-free versus no smoke-free rule households across specific categories of other demographics (e.g., income) or was just one chi-square test conducted comparing groupings? If the former, was any adjustment done for multiple testing? If the latter, the overall chi-square should likely not be interpreted as implying differences in one specific group across the grouping factor as communicated in lines 156-159.

Reviewer 2 Report

This study repeats well-known facts. Sampling methods applied in this study may lead to serious bias. The number of variables analyzed in this study is very limited. There is a limited number of statistics applied and the results are predictable. There is a lack of novel practical implications of this study. 

Findings from this study are unclear due to the potentially biased sample as well as a limited number of questions applied in this study. The authors should provide a more comprehensive questionnaire (detailed characteristics of household, detailed characteristics of smoking habits) as well as improve sampling methods.

Reviewer 3 Report

This study uses biomarkers to assess childhood exposures to secondhand smoke. Its a valuable study to the field, showing a high OR (large confidence intervals due to sample size) for childhood who live in a household with at least one smoker but a non-smoking mother AND especially with a smoking mother. 

1. Lines 46-53 - does NC (where study population comes from) have smoke-free home rules? Could clarify in methods when discussing population.

2. Table 3 should include N values for each predictor

3. Table 3 should include specifications of Model 1 and Model 2 as a footnote. 

4. The study population includes a wide range of children from ages 3-13 which suggests differences in behaviors and duration in exposure (for example line 212). Although I do recognize cotinine to reflect smoking during recent period of time (16-19 hours). Why didn't the Authors control for children's age in the models? 

5. Authors should note or comment on differences by race as notably differences in saliva cotinine have been reported for African American compared to White even at similar levels of smoking. cite  10.1158/1055-9965.EPI-16-0920

Reviewer 4 Report

This study presents findings from a prospective cohort study that examines the impact of smoke-free home rules on child secondhand smoke exposure. This is an interesting and important study that addresses the potential effectiveness of adding smoke-free rules at home and protect children from secondhand smoke. The central argument of the paper is clear, and the logical structure is well organized. However, there are a few comments that the authors should address.

    Introduction
    -This study was focusing on participants were “mother-child dyads”, however, the introduction should state any existing evidence built from this population but said “especially smoking mothers” at the end suddenly. The introduction was talking about the general findings of the effectiveness of smoke-free home rules on secondhand smoke, so why are authors addressing “mother-child dyads” relationship rather than include father or other family members in this study?

     -What does the literature say on the presence of smokers in the household? How does this study build upon that literature to inform the design (Model 1&2)?

     -What is meant by "using a cotinine threshold of ≥1 ng/ml"? At least authors should expand this part a little bit more or move it to Method part.

    Methods
    - What is meant by "limit of detection" for plasma nicotine concentrations? I would suggest adding more details for readers.

- For Home Smoking Rule, I guess the question was asking only for cigarette smoking. So how about other tobacco products such as cigar, e-cigarettes, are they also allowed or not allowed at home?

- Along with the introduction, for other covariates, other variables might need to be considered such as marital status, family members, paternal race, mother’s age, etc. In addition, is the household income from parents or only mother?

    Results
    -Should add more information on Model 1&2. Or describe more in the method part. Why only controlling household smoke status? For Model 2, even mother is non-smoker, comparing to no smoker at home, secondhand smoke exposure risk is significant. Does it mean other factors such as paternal factors may have an impact on the results?

    Discussion
    - Recall bias might be introduced since the study demonstrated cross-sectional effects   

    - “The majority of households (59.0%) in this study had a smoke-free home rule. Households with smokers had lower rates of smoke-free home rules, compared to households without smokers.” This selection bias might be directly bias the present results and findings. Suggest adding a sentence to give context to this or at least justify by future studies with possible study designs on how to avoid this.